# Pharmacokinetics and Childhood Obesity: Pathophysiological Basis and Challenges in Choosing the Ideal Body Size Descriptor

**DOI:** 10.3390/ph19010016

**Published:** 2025-12-21

**Authors:** Yolanda Hernández-Gago, José Germán Sánchez-Hernández, Pedro J. Alcalá Minagorre, Belén Rodríguez Marrodán, Laura Hernández Sabater, María José Cabañas Poy, Ana Cristina Rodríguez Negrín

**Affiliations:** 1Department of Hospital Pharmacy, Insular Maternal and Child University Hospital Complex, 35016 Las Palmas de Gran Canaria, Spain; arodnegz@gobiernodecanarias.org; 2Department of Hospital Pharmacy, Salamanca University Healthcare Complex, 37007 Salamanca, Spain; jgermansanchez@saludcastillayleon.es; 3Department of Pediatrics, Alicante Institute for Health and Biomedical Research (ISABIAL), Dr. Balmis University General Hospital, 03010 Alicante, Spain; alcala_ped@gva.es; 4Department of Hospital Pharmacy, Puerta de Hierro University Hospital, 28222 Madrid, Spain; belenrodma@gmail.com; 5Department of Pediatrics, Northwest Regional Hospital, 30400 Murcia, Spain; laurahdzsab@gmail.com; 6Department of Hospital Pharmacy, Hospital Universitario Vall d’Hebron, 08035 Barcelona, Spain; mjosep.cabanas@vallhebron.cat

**Keywords:** obesity, pediatric, size descriptor, pharmacokinetics, pharmacodynamics, pathophysiological change

## Abstract

Despite the progressive increase in obesity and associated chronic diseases in children, there is limited evidence on the optimal dosage of most medications for obese children and adolescents. This review analyzes the influence of pathophysiological changes on pharmacokinetics and pharmacodynamics and evaluates the body size descriptors used in clinical practice. Patients with obesity present significant pathophysiological alterations, such as a substantial increase in fat/lean mass ratio, increased blood flow and cardiac output, and changes in plasma protein binding, which may affect the volume of distribution of drugs and the adjustment of the loading dose. In these patients, the distribution volume of hydrophilic drugs appears to slightly increase, while it varies widely—depending on the drug and other factors such as affinity for other tissues—for lipophilic drugs. On the other hand, a reduction in tissue perfusion, alterations to liver enzyme activity, and an increase in liver and kidney mass and blood flow have been reported, indicating a possible modification in drug clearance and necessitating adjustments to maintenance regimens. Furthermore, while there are multiple size descriptors, it is difficult to establish a single dosing strategy for the obese population, given the lack of studies confirming the extent of changes in pharmacokinetic processes, which will also depend on the properties of each drug, such as liposolubility and elimination pathways. New strategies need to be developed to characterize pharmacokinetic and pharmacodynamic changes in the obese pediatric population in order to optimize dosing regimens and improve the safety and efficacy of treatments.

## 1. Introduction

Obesity and overweight are serious public health problems at all ages [1]. In children and adolescents, the prevalence of obesity has increased dramatically and progressively over the last several decades in all regions of the world, and it is now the most prevalent form of malnutrition worldwide in both sexes, ahead of underweight [2]. According to the World Health Organization (WHO), rates of overweight and obesity continue to increase in adults and children. From 1990 to 2022, the percentage of children and adolescents aged 5 to 19 years living with obesity increased fourfold globally, rising from 2% to 8%. In 2022, more than 390 million children and adolescents aged 5 to 19 years were classified as overweight (20%), of whom 160 million (8%) were classified as obese [3]. Obesity is expected to continue to increase across populations and regions, reaching estimates of 15.6% among children aged 5–14 years by 2030 and 14.2% among adolescents aged 15–24 years by 2050 [4].

Excess body weight in children and adolescents significantly increases the risk of developing chronic diseases that affect most organs and systems [5]. On the other hand, an increased risk of secondary obesity/overweight has been described in children with chronic diseases, such as those with musculoskeletal disorders or those who are cancer survivors [6]. Together, these factors contribute to an increase in healthcare demands and the use of pharmacotherapeutic resources [7] and a higher risk of adverse effects linked to healthcare [8], such as those related to inadequate drug dosing [9].

During the different phases of physiological growth in childhood, changes occur in body composition, organ function, and maturation. These variations contribute to differences in pharmacokinetic parameters in children compared to those in the adult population, leading to changes in drug disposition and the need to adapt dosage regimens [10]. Excess body weight at this age is associated with additional pathophysiological changes that can influence the pharmacokinetics (PK) and pharmacodynamics (PD) of some drugs [11]. These changes may lead to specific dosing requirements in this population, aimed at preventing toxicity or therapeutic failure [12].

In pediatrics, most prescriptions are based on total body weight (TBW), assuming that the patient has a normal weight and height for their age. For obese children, prescribing based on TBW can lead to inappropriate treatment guidelines; dosages may even exceed the recommended doses for adults, which can have significant clinical implications [13]. However, the use of other dose adjustments, such as ideal body weight (IBW) or adjusted body weight (AjBW), could lead to underdosing and therapeutic failure in other cases [11,13,14].

In this group of patients, the application of dosage adjustment mechanisms is not systematized, and no single optimal body size descriptor has been established for most commonly used drugs. In recent decades, several reviews have been published on dose individualization for obese patients, highlighting the lack of solid scientific evidence [11,12,14,15,16,17,18,19,20,21,22,23]. In this context, it is worth noting that in a systematic review conducted to determine changes in drug disposition in obesity, information was only found for 21 drugs, highlighting that most of the studies included a very small number of patients and a wide variability in dosing strategies, as well as in PK parameters [23]. In another review aimed at determining dosing for drugs commonly used in children and adolescents, information was only found for 16 of the 70 drugs studied [14]. The absence of specific recommendations for dose adjustment in overweight pediatric populations creates a potential safety risk, especially with high-risk medications or in the treatment of serious diseases. Similarly, a systematic review assessing dosing information for twenty-five drugs commonly used in intensive care settings reported that pediatric information was only included on the FDA label for two of them, and for neither of them were sufficient PK data available [18].

Moreover, obese children exhibit distinct alterations in body composition that differ from those observed in adult obesity, together with ongoing maturational processes affecting enzymatic activity, renal function, and hemodynamics. These effects imply that direct extrapolation from adult data or the use of simple weight-based descriptors may lead to potentially unsafe dosing. A review of PK studies in obese patients specifically addressed the prediction of adolescent dosing based on adult data and concluded that such extrapolation is not appropriate due to the limited number of available studies and the contradictory results reported [24]. Another study reached similar conclusions after comparing PK behavior, with a particular focus on clearance (Cl), for midazolam, busulfan, and metformin, showing that patterns in Cl variation in obese adults do not mirror those observed in obese children [25].

Given the limited and heterogeneous nature of the available evidence, clinical practice often relies on heuristic dosing rules that do not consistently ensure patient safety. Therefore, new approaches to dose optimization in obese pediatric patients should be proposed, particularly for drugs for which the strength of evidence is low. A key starting point is an improved understanding of the PK/PD changes resulting from obesity-related pathophysiological alterations in children. When combined with drug-specific characteristics—such as molecular size, lipophilicity, plasma protein binding, hepatic extraction, and other relevant properties—this knowledge may help inform more appropriate dosing strategies in this population. Accordingly, the objective of this review is to analyze the influence of pathophysiological changes associated with childhood and adolescent overweight and obesity on the pharmacokinetic and pharmacodynamic parameters of drugs, as well as to summarize and evaluate the body size descriptors used in clinical practice to optimize dosing in this population.

## 2. Conceptualization of Obesity and Overweight in Pediatric Patients

Obesity and overweight are pathological conditions secondary to excess body fat that manifest as excess weight and body volume and can produce potential metabolic, physical, and psychological manifestations [26]. These conditions are usually diagnosed using indirect indicators of body fat through simple anthropometric measurements such as body mass index (BMI). This weight-to-height ratio varies physiologically throughout different pediatric ages, and diagnosis requires the use of standard growth charts according to age and sex to detect pathological deviations in body weight in children and adolescents [27]. Although this approach has limitations, the relationship between the patient’s BMI and these charts is the most widely used indicator in clinical practice. This methodology allows for the simple quantification of excess body weight according to deviation from statistical normality and the resulting stratification of patients into the categories of overweight and different degrees of obesity (mild, moderate, and severe) [28].

There is some variation in the methodological establishment of cut-off points for the operational definition of obesity and overweight in children and adolescents. The WHO recommends using the weight-for-height z-score to diagnose overweight and obesity in children under 5 years of age, based on growth patterns developed in 2006, which include data from 8500 children from across diverse ethnic backgrounds and cultural contexts [29]. It defines excess weight based on standard deviations from the weight-to-height ratio, defining overweight as greater than two standard deviations (z-score + 2) and obesity as greater than three standard deviations (z-score + 3) above the median. For children aged 5 to 19 years, based on standard deviations from BMI for age and sex, overweight is defined as greater than one standard deviation (z-score + 1) and obesity as greater than two standard deviations (z-score + 2) above the median of the WHO growth patterns [3], as detailed in Table 1 [3,7,22].

The International Obesity Taskforce, the American Academy of Pediatrics, and the Centers for Disease Control and Prevention (CDC) recommend using BMI based on sex and age as a reference due to changes in weight and height ratios throughout development and growth, according to CDC growth charts. Thus, a patient is defined as overweight when the BMI percentile is ≥85th percentile and obese when the percentile is ≥95th percentile for a given age and sex in children over 2 years of age, as shown in Table 2 [30,31,32].

The definition of obesity or overweight in children under 2 years of age is difficult to categorize using classic anthropometric measurements [33]. Although the WHO recommends a high weight-to-length ratio in children under 5 years of age, an elevated BMI in infants has been identified as the best predictor of obesity in the later stages of childhood [34]. However, it has been pointed out that BMI has limitations [35] in assessing excess weight and adiposity in infants, partly due to the unique changes in body composition that occur physiologically during this pediatric stage [36].

## 3. Pathophysiological Changes in Obese Pediatric Patients

Obese children have a significant excess of body mass, in terms of both fat mass and other components such as lean mass and bone mineral content [37], with the increase in fat mass being substantially greater. It is generally suggested that 75% of excess weight corresponds to fat mass [38], which represents 30% to 50% of total body weight [22]. Half of the excess fat mass accumulates in the trunk, one third in the legs, and the rest in the arms. Lean mass is distributed more evenly, while excess bone mineral content is concentrated in the legs. Overhydration of lean mass has also been observed, which is attributed to an increase in extracellular volume [37]. Figure 1 details the alterations in body composition associated with pediatric obesity.

There are other notable changes in body composition and systemic blood circulation. In general, obese pediatric patients tend to be taller than those of normal weight for the same age and sex [37] and have a relative increase in blood volume and cardiac output [39], which correlates better with lean body weight (LBW) and body surface area (BSA) than with excess fat mass [40]. On the other hand, blood flow to adipose tissue has been shown to be lower in obese patients, although it increases in the postprandial state, possibly associated with insulin resistance and increased vascular resistance. This could limit the distribution of lipophilic drugs [11,12,20,22]. A reduction in tissue perfusion has also been described in obese patients, as well as an increase in hepatic mass and blood flow and an increase in renal mass, potentially altering the function of the excretory organs [30,38,41].

The spectrum of comorbidities affecting overweight and obese pediatric patients is very broad and affects numerous systems, including prediabetes and diabetes mellitus with insulin resistance, dyslipidemia, high blood pressure, metabolic syndrome, polycystic ovary syndrome, precocious puberty, gastroesophageal reflux, respiratory complications such as asthma and obstructive sleep apnea, orthopedic diseases, and psychiatric disorders. Many of these pathological conditions interact synergistically with each other and can lead to alterations in organs that are key to the metabolism of many drugs [7,42].

Up to 90% of obese patients present histological liver abnormalities related to fat infiltration, such as non-alcoholic fatty liver disease (NAFLD). In the pediatric population, up to 40% of obese children and adolescents meet the criteria for metabolic dysfunction-associated fatty liver disease with steatosis, including non-alcoholic steatohepatitis (NASH) [43], with varying degrees of progressive and chronic inflammation, which can lead to cell damage, fibrosis, and impaired liver function. It has been shown that these liver changes modify drug transporters and hepatic elimination and increase the likelihood of drug interactions [44]. Excess body weight in childhood and adolescence also generates a proinflammatory state [45] with an increase in inflammatory cytokines such as interleukin 6 and C-reactive protein, which can alter the regulation of cytochrome P450 enzymes and hepatic drug transporters, as well as affecting the homeostasis of innate and adaptive immunity [46].

There is evidence to suggest that overweight and obesity are also associated with adverse effects on renal function in children [47]. Obese children face a higher risk of kidney damage compared to their non-obese peers, characterized by a progressive decline in renal function and the development of proteinuria [48]. An early feature of obesity-related kidney damage is glomerular hyperfiltration, leading to rates exceeding physiological levels. This condition is closely related to an increase in fat mass, which increases metabolic demands and cardiac output, leading to increased renal blood flow that may influence the elimination of a large number of drugs [49].

## 4. Impact of Pathophysiological Changes on Pharmacokinetic and Pharmacodynamic Processes

The pharmacokinetic parameters most affected by obesity are the apparent volume of distribution (Vd) and Cl. The magnitude of the effect depends, among other factors, on the fat solubility of the drug, the degree of plasma protein binding, and tissue perfusion [11,41]. Vd, measured in liters, is a theoretical PK parameter that establishes the relationship between the total amount of drug administered and the measured plasma concentration, providing an estimate of the extent of tissue distribution. Cl, measured in L/hour or mL/minute, represents the volume of plasma from which the drug is completely eliminated per unit of time and can be conceptualized as the sum of individual organ clearances, primarily those of the liver, kidneys, and lungs [11,50]. Adequate characterization of Vd and Cl in the obese or overweight pediatric population is essential in rationally defining loading/bolus doses (determined mainly by Vd) and maintenance doses (determined by Cl) in dosing regimens, thereby optimizing exposure and achieving PK/PD objectives [11,41].

The following paragraphs describe the PK changes in the processes of absorption, distribution, metabolism, and excretion (ADME), as well as the most relevant PD changes in obese pediatric patients. Table 3 summarizes the pathophysiological changes and how they can affect PK parameters, and Figure 2 shows physiological and PK/PD alterations in pediatric obesity.

### 4.1. Absorption

Absorption in the gastrointestinal tract determines the amount of drug that reaches the bloodstream, as well as the rate constant (ka) and time (Tmax) at which the maximum concentration (Cmax) is reached. These parameters depend on the physicochemical properties of the drug, pH, permeability, gastric emptying rate, regional blood flow, first-pass effect, and intestinal flora [16].

Although data are very limited in obese adult patients, an increase in intestinal permeability and splanchnic blood flow and accelerated gastric emptying have been observed, resulting from an increased body surface area, cardiac output, and microbiota composition. This could lead to faster absorption, although the percentage absorbed would be the same as in non-obese patients. Based on the available studies, no clear clinical relevance has been demonstrated; nevertheless, potential changes in bioavailability and absorption rate cannot be excluded. Moreover, it should be acknowledged that the time to reach Cmax may be clinically relevant for drugs requiring a rapid onset of action, including analgesics, anesthetics, and antiepileptic agents. To date, the available evidence is insufficient to support clinical recommendations regarding absorption-based dose adjustments in obese children [16,25,41,51].

Regarding the absorption of inhalation anesthetics in obese adult patients such as enflurane, sevoflurane, or halothane, faster induction of anesthesia has been demonstrated despite increased cardiac output, which is likely attributable to a reduction in blood/gas partition [16]. In relation to absorption via other routes, such as subcutaneous, intramuscular, or rectal administration, the available evidence is even more limited. Studies assessing intramuscular epinephrine in pediatric patients with anaphylaxis suggest that the amount of drug absorbed may be compromised due to the needle length in auto-injectors. In obese patients, the increased skin-to-muscle distance may result in unintentional subcutaneous rather than intramuscular administration, thereby delaying the onset of action and reducing the expected Cmax values [54,55]. With regard to subcutaneous administration, the risk of inadvertent intramuscular injection is low in obese children; however, appropriate needle length and correct injection technique should be ensured. A study involving obese pediatric and adult patients with primary immunodeficiency receiving subcutaneous immunoglobulin has shown consistent absorption and comparable bioavailability between obese and non-obese patients [56]. The authors have not identified any published pharmacokinetic studies on rectally administered drugs in obese pediatric patients, nor in animal models or adults, that would enable the extrapolation or identification of trends to inform clinical practice.

### 4.2. Distribution

#### 4.2.1. The Apparent Volume of Distribution

The apparent volume of distribution describes the extent to which a drug is distributed from the vascular compartment to the body’s tissues and is a key parameter for interpreting the relationship between the plasma concentration and the total amount of drug in the body. It is also decisive in estimating the loading dose required when a specific therapeutic concentration is to be reached quickly. Vd depends on the physicochemical properties of the drug, such as fat solubility, molecular size, and degree of ionization, as well as the degree of plasma protein binding, tissue perfusion, and plasma partition coefficients in tissues [24,50].

In obese children, Vd may be altered due to increased circulating blood volume, increased extracellular water, decreased tissue perfusion, and an increased ratio of fat mass to lean mass [16,30,57]. Relatively hydrophilic drugs that are distributed in the blood space and extracellular water are expected to show a slight increase in Vd that correlates with LBW. Lipophilic drugs with extensive diffusion in fat will show a higher Vd in obese patients that correlates with TBW [30,38,58].

However, the impact of obesity on the Vd of drugs can vary widely, as blood flow to the adipose tissue is low, and increased fat does not equate to increased effective perfusion. Therefore, the effect will depend largely on the type of drug. Above all, it has been found that there is no clear evidence of an increase in Vd with highly lipophilic drugs; the in vivo distribution may not correlate directly with an increase in adipose tissue, as these drugs have a high affinity for other tissues. For example, digoxin, procainamide, glyburide, prednisolone, and methylprednisolone have low affinity for adipose tissue and higher partition coefficients in other tissues. Digoxin has a high affinity for skeletal muscle, and glyburide has a high affinity for well-perfused organs such as the heart, kidneys, and liver. These highly lipophilic drugs do not show the expected increase in Vd because they are distributed in organs where there is no significant difference in volume between obese and non-obese patients [24,51]. In general, there appears to be a slight increase in Vd for hydrophilic drugs, while for lipophilic drugs, it varies significantly depending on the drug, as well as on other determining factors such as plasma protein binding, the degree of ionization of the drug, affinity for other tissues, and patient comorbidities [24].

From a clinical point of view, the dosing strategy should be aimed at achieving therapeutic concentrations without compromising safety. For hydrophilic drugs with limited binding to adipose tissue, such as neuromuscular blocking agents and aminoglycosides, the use of TBW may lead to drug overexposure; therefore, dosing should be based on IBW or on an adjusted body weight reflecting a percentage of excess body weight over IBW. For lipophilic drugs, such as clarithromycin and midazolam, for which increased pharmacokinetic variability has been described, it may be reasonable to use a loading dose based on TBW, administered in divided or staggered doses or accompanied by close monitoring of clinical response and/or plasma concentrations, particularly for drugs with a narrow therapeutic index. However, given that the distribution patterns—especially in lipophilic drugs—can be unpredictable, therapeutic drug monitoring (TDM) should be considered whenever feasible. This is particularly relevant for drugs with a narrow therapeutic window, such as aminoglycosides, vancomycin, antiepileptic drugs, and digoxin, among others [15,38,58].

#### 4.2.2. Plasma Protein Binding

Many drugs circulate in the bloodstream partially bound to plasma proteins, mainly albumin and α1-acid glycoprotein. The free fraction determines the pharmacologically active portion and, therefore, can modify both Vd and clearance at steady state.

Regarding albumin—the main protein to which acidic drugs bind—no significant change has been observed in obese patients, whereas with α1-acid glycoprotein —the main transporter of basic drugs—an approximately twofold increase has been determined in obese adult patients. To date, this increase has not been reproduced in children due to a lack of studies in this population [16,30,41,50]. It is also to be expected that obese patients have higher levels of triglycerides and cholesterol, which may affect the binding of drugs to lipoproteins; however, it remains unclear how this alteration could affect Vd [16].

### 4.3. Metabolism

The goal of drug metabolism is to produce an increased number of hydrophilic metabolites that are more easily eliminated from the body. Biotransformation occurs mainly in the liver and involves phase I reactions in which there is some functional alteration and phase II reactions or conjugation with endogenous substances (such as glucuronidation, sulfation, methylation, or acetylation) that convert the drug into more polar products and facilitate its elimination by the kidney, bile, and/or feces. Other organs with significant metabolic capacity include the gastrointestinal tract, kidneys, and lungs [50].

With regard to estimating Cl in obese patients, a key PK parameter for designing maintenance dosage guidelines, there is less clarity. According to some authors, Cl should be determined mainly by LBW, since lean tissue is reported to be highly correlated with drug metabolism, and adipose tissue is believed not to contribute to metabolic processes; thus, maintenance doses generally do not increase proportionally with the patient’s weight but rather with LBW. It should be taken into account that this assertion is largely based on theoretical pharmacokinetic considerations and that several studies have reported conflicting data. According to one review, LBW was the size descriptor associated with Cl in only 5 of the 15 drugs evaluated, whereas TBW was identified as the relevant descriptor for CL in the remaining studies. From a clinical perspective, this distinction is highly relevant, as it directly determines the maintenance dosing required to achieve and maintain therapeutic concentrations at steady state [24,41,51]. Moreover, it is important to consider that pathological alterations such as fatty liver and/or hepatic steatosis and the presence of inflammatory cytokines can alter enzymatic reactions in the liver, hepatic blood flow, and the activity of hepatic transporters [24,25,30,52,59].

In general, Cl will depend on the hepatic metabolic capacity, blood flow, perfusion of organs such as the liver and kidneys, and hepatic extraction rate of the drug. In drugs with low or intermediate hepatic extraction, Cl will mainly be determined based on plasma protein binding and the expression and activity of metabolizing enzymes, while in drugs with high hepatic extraction, tissue perfusion will be the rate-limiting factor for Cl [25].

#### 4.3.1. Phase I Metabolism

Phase I reactions (oxidation, reduction, and hydrolysis) are mainly catalyzed by the cytochrome P450 system, whose function is to introduce or expose polar functional groups to facilitate phase II metabolism or the direct excretion of the drug. These transformations are mainly catalyzed by the cytochrome P450 system located in the hepatic endoplasmic reticulum, although other organs—such as the intestine, kidneys, and lungs—also participate to a lesser extent [50,52].

The changes in cytochrome P450 enzyme activity in obese versus non-obese patients are detailed below.

##### CYP3A4

The CYP3A4 enzyme is the main metabolizer of phase I reactions, responsible for the biotransformation of approximately 50% of drugs. Studies conducted in obese adults compared to non-obese adults have shown a 40% decrease in activity, and body weight-normalized clearance values show that drug Cl per kg of body weight is reduced by approximately half in obese individuals [30,51,52].

There are no studies that reproduce these data in children and adolescents, with contradictory data found in some cases. Studies with clindamycin and midazolam detailed a decrease in normalized Cl in children [60]. Meanwhile, a similar or slightly elevated Cl has been described in obese children, in one study with fentanyl and two with midazolam [30,61,62,63]. These results may be due to possible compensation based on increased liver size and hepatic blood flow in drug clearance, as well as affinities for different enzymes of the CYP3A complex [24,30].

##### CYP2E1

The CYP2E1 enzyme accounts for 5% of the metabolizing activity of phase I reactions. In adults, a 140% increase in activity has been observed, correlated with total weight and the degree of steatosis. When Cl is normalized by weight, Cl in obese patients is comparable to that in non-obese patients, implying that CYP2E1 enzyme activity increases proportionally with weight [52]. A study with chlorzoxazone in pediatric patients suggests an increase in the activity and/or expression of this enzyme, leading to an absolute Cl that is twice as high [64].

The increased activity of CYP2E1 in obese patients calls for caution in the use of paracetamol, as this enzyme forms the toxic metabolite NAPQI. However, 90% of the drug is metabolized in phase II and only 5–10% by CYP2E1, and, to date, evidence has not clearly demonstrated an increased risk of acetaminophen-induced liver injury in obese children and adolescents. Hepatotoxicity may depend on the balance between the rate of formation of the metabolite NAPQI by CYP2E1, the capacity of acetaminophen elimination through the glucuronide and sulfate conjugation pathways, and the maximal rate of hepatic glutathione synthesis, which is responsible for neutralizing NAPQI [65,66,67], so further studies are needed to clarify the role of CYP2E1 in paracetamol toxicity in obese adults and children [25,51,52].

##### Other Cytochrome CYP450 Enzymes

There are other enzymes that show variations in their metabolic activity in the context of obesity, although the results are heterogeneous and, in some cases, dependent on genetic factors such as CYP2D6 and CYP2C19 enzymes. Studies conducted with these enzymes in obese adult patients show a tendency toward increased enzymatic activity. Studies conducted in obese adult patients with the CYP2C9 enzyme also point to an increase in activity with phenytoin and ibuprofen [51,52].

The CYP1A2 enzyme shows an inducing effect in patients who smoke. A slight increase in its activity was observed in obese adult patients, while a decrease in activity was observed in children treated with caffeine, although this was not statistically significant [52,68].

##### Xanthine Oxidase: Another Enzyme Involved in Phase I Metabolism

A pediatric study with caffeine showed a 16% increase in xanthine oxidase activity in obese children compared to non-obese children [52,68]. Another study conducted in children with acute lymphoid leukemia treated with mercaptopurine showed an increase in clearance in overweight and obese children and a correlation between body fat mass and the area under the curve (AUC) achieved using mercaptopurine [69].

#### 4.3.2. Phase II Metabolism

##### Uridine Diphosphate Glucuronosyltransferase (UGT)

The UGT complex includes two families of enzymes, UGT1 and UGT2, and three subfamilies, UGT1A, UGT2A, and UGT2B. Many of the enzymes in the UGT complex are expressed not only in the liver but also in the gastrointestinal tract, adipose tissue, and kidneys, where glucuronidation can be important, for example, in the metabolism of drugs such as paracetamol, garenoxacin, oxazepam, and lorazepam. Studies with oxazepam and lorazepam have also shown higher Cl values in obese adults compared with non-obese individuals [25,52].

Paracetamol is extensively metabolized by the enzymes of the UGT complex in adults and children, and a higher Cl has been observed in obese versus non-obese individuals. A study conducted in adolescents with NAFLD found no differences between weight-normalized Cl in obese and non-obese individuals, apart from a significantly higher concentration of conjugated paracetamol in urine compared to unconjugated paracetamol, indicating an increase in metabolism mediated by the UGT enzyme complex, which leads to higher clearance in absolute terms in obese adolescents [52,70].

In general, we can state that there is a significant increase in Cl in obese patients compared to non-obese patients and, consequently, weight-normalized Cl is equal to or slightly lower in obese patients than in non-obese patients [52].

#### 4.3.3. Other Phase II Metabolic Enzymes

It should be noted that, among other phase II metabolic enzymes, N-acetyltransferases are particularly relevant, accounting for approximately 5% of conjugation reactions, together with the S-glutathione transferase enzymes.

A study conducted in obese adult patients with procainamide showed that plasma Cl due to acetylation was slightly higher in obese patients, although this increase was not significant. In a pediatric study, N-acetylation of caffeine was five times higher in obese patients in the slow acetylator genotype group [52,68]. On the other hand a higher Cl has also been observed in obese patients with busulfan, which is metabolized by the enzyme S-glutathione transferase [52]. However, a pediatric study to determine whether BMI influences PK parameters concluded that there are significant differences in the doses required by non-obese and obese patients, as well as in Cl and AUC. Obese pediatric patients had lower Cl, reaching a significantly higher AUC [52,71].

In summary, changes in hepatic metabolism vary depending on the enzyme complex affected. See Table 3 for details on the direction of change.

#### 4.3.4. Hepatic Blood Flow

Hepatic blood flow is increased in obese patients due to increased cardiac output and increased blood flow and liver size, although it may be reduced in patients who have developed NASH due to the narrowing of the hepatic sinusoids [30].

Alterations in hepatic blood flow mainly affect drugs with a high degree of hepatic extraction (>1.5 L/min). There are very few studies involving this type of drug, so it is very difficult to draw conclusions about how it may affect Cl, and there have been no studies conducted in pediatric populations [51,52]. Studies performed in adult populations treated with fentanyl, propofol, propranolol, and labetalol suggest that hepatic blood flow is likely to be increased in obese patients [16,52].

### 4.4. Excretion

The kidney is the main organ responsible for the elimination of drugs and involves three distinct processes: glomerular filtration, active tubular secretion, and passive tubular reabsorption. Other routes of elimination include pulmonary excretion, mainly for anesthetic gases; fecal elimination, for drugs that are administered orally and are not absorbed; and other routes such as sweat, saliva, and tears, which are quantitatively less important [50].

#### 4.4.1. Glomerular Filtration

The amount of drugs eliminated by glomerular filtration depends on the filtration rate and plasma protein binding, as only the free fraction is eliminated. Increases in Cl have been reported in obese adult patients with vancomycin, aminoglycosides, daptomycin, carboplatin, and low-molecular-weight heparins due to a higher glomerular filtration rate [25,52].

Studies examining vancomycin in pediatric contexts have established an increase in absolute Cl and a decrease in weight-normalized clearance, supporting the use of allometric weight with an exponent of 0.75 for calculating Cl [30,72].

It should be taken into account that obesity alters the relationship between muscle mass and serum creatinine, which may bias glomerular filtration rate estimates. Thus, the Schwartz formula tends to overestimate renal function in overweight and obese children because—although serum creatinine may be within the reference range—the relationship between muscle mass and body weight is altered in these patients. Comparative evidence in obese children shows that the FAS (full-age-spectrum) equations by age or height and LMR18 (revised and adjusted Lund–Malmö) are more accurate and less biased than the bedside CKiD and Schwartz formula [73,74,75]. On the other hand, it is recommended that glomerular filtration be indexed based on body surface area calculated with IBW, since TBW can lead to its overestimation [76].

#### 4.4.2. Tubular Secretion

Active tubular secretion is mediated by renal transporters. Studies with procainamide, ciprofloxacin, cisplatin, topotecan, and digoxin have been conducted in adult patients. In all the studies reviewed, there was a tendency toward increased Cl due to increased tubular secretion. Weight-normalized Cl was found to be equal or slightly lower in obese patients compared with non-obese patients [30,52].

#### 4.4.3. Tubular Reabsorption

Passive tubular reabsorption occurs when the drug is reabsorbed from the tubular lumen into the systemic circulation. There are few studies describing obesity-mediated alterations in the tubular reabsorption of drugs. A study conducted in adult patients taking lithium reported a decrease in tubular reabsorption, resulting in significantly increased Cl in obese patients [30,52]. On the other hand, it has been reported that tubular sodium reabsorption is higher due to glomerular hyperfiltration [52].

Scientific evidence regarding tubular reabsorption in obese children is derived from physiological models and simulations rather than direct clinical trials and suggests that it may be altered, but the magnitude and direction of this effect depend on the characteristics of the drug and the degree of obesity [53].

In summary, Cl levels from renal excretion are higher in obese individuals due to an increase in glomerular filtration and tubular secretion, with no conclusive studies regarding tubular reabsorption.

### 4.5. Pharmacodynamic Alterations Associated with Obesity

In addition to pharmacokinetic changes, obesity in pediatrics can alter the PD of drugs, modifying receptor sensitivity, the tissue density of therapeutic targets, and the concentration–effect relationship.

There are very few studies supporting an alteration in response to drugs in obese pediatric patients. One such study, Hanafy et al. concluded that obesity reduces the efficacy of calcium channel antagonists in children with kidney disease, especially in lowering systolic blood pressure. In adults, greater sensitivity to glipizide-type hypoglycemic agents has also been demonstrated [38,77].

On the other hand, it is known that obesity is related to a proinflammatory state that can alter the homeostasis of the immune system. Thus, responses to treatments that include immune modulation may be altered. Obese asthmatic patients show resistance to treatment due to higher levels of IgE and eosinophilia, and, therefore, they may require higher doses of omalizumab. In contrast, no adjustment to the immunoglobulin dose has been described as necessary in patients with primary immunodeficiency syndrome. The response to immunization with some vaccines, such as the flu vaccine, is also altered, with an increase in the immunogenic response, while no change has been observed in the response to hepatitis B or tetanus vaccines. Likewise, obesity has been associated with an altered immune response to various infections, including viral, cutaneous, nosocomial, and surgical infections [16,25,78].

Obesity has been identified as an independent risk factor associated with a lower therapeutic response and lower remission rates in obese patients treated with biological drugs, because the chronic inflammation triggered by obesity can cause changes in clearance related to proteolysis that increase with body weight, causing a higher probability of neutralization by antibodies [16,30].

Insulin sensitivity is altered in these patients due to the occurrence of chronic inflammation, which can compromise the response to exogenous insulin and often requires dose escalation, especially in patients with type 2 diabetes [16].

However, no studies have yet been adequately designed to discern possible changes in response in obese patients.

## 5. Body Size and Composition Descriptors

Key questions remain unresolved regarding which methodology to use to calculate dosage in overweight pediatric populations. Drug dosing in pediatric patients is determined by physiological developmental characteristics at different stages of pediatric development. Thus, pediatric dosing guidelines include dosage recommendations by age group, considering that PK changes secondarily to growth, functional maturation, and body composition. The main PK changes occur during childhood, and it is considered that renal function reaches maturity at 2 years of age and that the main enzymatic systems in the liver responsible for the biotransformation of drugs mature after the first year of life [58,79].

The subgroups that may be of interest regarding obese patients are early childhood (2–5 years), middle childhood (school age, 6–11 years), and early adolescence (12–18 years) [30,80].

Many body size descriptors have been utilized for scaling doses in pediatrics, and they can sometimes differ between the calculation of the loading dose and that of the maintenance dose. Regardless of the descriptor used, the maximum dose (dose capping) must always be considered. If this is not established in pediatrics, doses should not exceed the doses recommended for adult patients and, in certain cases, those for obese adults [15,24,30,57].

We will now discuss the particularities of the main descriptors used in overweight and obese children and adolescents, with the methodology detailed in Table 4.

### 5.1. Total Body Weight

Dosing based on TBW is the most widespread method for calculating doses in pediatrics, although fixed dosages based on age and/or weight ranges may sometimes be used—mainly for drugs with a wide therapeutic margin, as this can lead to the overestimation of the actual dose required. In obese children, TBW has been proposed as a descriptor for the dosage of various antimicrobials, such as cephalosporins and penicillins, mainly due to their hydrophilic properties, renal elimination, and wide therapeutic range and the potential risk of underdosing [12,14,15,22,30,57,58].

### 5.2. Body Surface Area

Dosage based on body surface area is considered for certain medications such as chemotherapy, immunotherapy, monoclonal antibodies, and some antiretrovirals, as well as for transfusion therapy and fluids, as it is assumed to correlate better with metabolism and cardiac output than TBW [91]. There are different formulas for calculating BSA, with the Haycock formula being the most accurate and validated and covering a wider age range [81]. The Mosteller formula is widely used because of its simplicity, but it tends to underestimate BSA in neonates and infants [82,92].

The dosing of blood products and other intravenous fluids based on BSA or the conversion of weight to BSA, especially in infants and children with excess body weight, requires further validation through clinical trials to establish its safety in various clinical settings [93,94].

### 5.3. Ideal Body Weight

Ideal body weight is a concept that cannot be measured for a specific patient and represents the weight associated with the highest life expectancy for a given age, sex, and height. In pediatrics, there are different methods of calculating IBW, such as the McLaren method, which was the first such method published (in 1972) and was originally developed to classify malnutrition. This method, like the one developed by Moore, is based on a growth chart and takes into account weight, height, and age. The Traub method was developed later to avoid dependence on growth charts. This method only requires the child’s height in centimeters [85]. Table 4 details Lexicomp’s simplified formula, which is applicable to children < 18 years of age with a height < 152 cm [83,84].

The methods described above are tailored to certain age subgroups, so the reverse BMI method based on BMI_50_ and height is often preferred for calculating IBW, as it can be applied to children aged 2 to 20 years. BMI_50_ can be calculated based on pediatric growth charts based on age and sex. However, calculating BMI_50_ using growth curves can be tedious and prone to error, so alternative methods have been developed, such as the nomogram and formulas developed by Callaghan and Walker for pediatric patients older than 5 years [86]:$${BMI}_{50}\;(girls)\;=22.82-\;\frac{7.51}{1+{\left(\frac{age}{13.46}\right)}^{4.44}}$$$${BMI}_{50}\;(boys)=24.27-\frac{8.91}{1+{\left(\frac{age}{15.78}\right)}^{4.40}}$$

IBW is generally used for hydrophilic drugs with a low body distribution, such as acyclovir. It has also been proposed for use with hydrophilic drugs where there is a high risk of serious effects from overdose, such as morphine, and for titration [15,17,22].

### 5.4. Adjusted Body Weight

Adjusted body weight is another intermediate size descriptor between TBW and IBW that takes into account lean weight plus a proportion of excess weight where the drug is estimated to be distributed. The correction factor applied can vary between a minimum of 0.25, which corresponds to a 25% increase in LBW in obese patients, and 0.4 [15,16]. AjBW is a body size descriptor widely used for aminoglycoside drugs with a factor of 0.35–0.4, as these are water-soluble drugs with extensive distribution in extracellular water [14,15,17,22].

### 5.5. Lean Body Weight and Fat-Free Mass

Lean body weight and fat-free mass (FFM) are used to estimate lean weight and the weight of vital organs, extracellular fluid, bones, and muscles. The difference between these descriptors is that LBW also includes fat accumulated in cell membranes, bone marrow, and the central nervous system. For practical purposes, they are considered interchangeable descriptors.

The calculation of LBW in pediatrics is based on the assumption that extracellular fluid volume is proportional to LBW according to the equation in Table 4. However, there are simplified calculations that assume that approximately 29% of excess weight in obese children is due to lean weight.

The gold standard for calculating FFM is dual-energy X-ray absorptiometry; however, in daily clinical practice, it can be calculated using the formulas described below, which were developed in a study including children and young adults aged 3–29 years based on sex, height, and weight [87,88].$$FFM\;(girls)=\left[1.11+\left(\frac{\left(1-1.11\right)}{\left[1+{\left(\frac{\mathrm{Age}}{7.1}\right)}^{-1.1}\right]}\right)\right]\times \left[\frac{\left(9270\times \mathrm{TBW}\right)}{8780\;+\left(244\times \mathrm{BMI}\right)}\right]$$$$FFM\;(boys)=\left[0.88+\left(\frac{\left(1-0.88\right)}{\left[1+{\left(\frac{\mathrm{Age}}{13.4}\right)}^{-12.7}\right]}\right)\right]\times \left[\frac{\left(9270\times \mathrm{TBW}\right)}{6680\;+\left(216\times \mathrm{BMI}\right)}\right]$$

There are fewer recommendations for the use of FFM as a dose descriptor in pediatrics. Two studies conducted with enoxaparin led to the conclusion that the use of FFM as a basis for dosing could represent the best strategy for equalizing exposure in children with and without obesity [95,96].

It is estimated that both Cl and the glomerular filtration rate are linearly related to LBW and FFM; however, it has been found that a nonlinear function using allometric scaling, which we will describe below, is more accurate and covers a wider body mass range.

### 5.6. Allometric Methods

Another approach to dosing in pediatrics involves the application of the principles of allometry. Allometric scaling is a technique used in quantitative biology that describes the relationship between body size and physiological and structural functions. The relationship that has been established between the basal metabolic rate of different animal species and body mass is a logarithmic slope with an exponent of 0.75, known as Kleiber’s law. This theory is based on West’s model, which assumes that the energy consumed per cell is similar in all organisms, but larger organisms have more support structures (bone, connective tissue) with a lower density of metabolically active cells, so total metabolism does not increase proportionally to weight but, instead, based on an allometric exponent of ¾ (0.75), which reflects the relationship between structure and biological function [89,97].

This methodology assumes that different physiological processes that influence drug disposition, such as blood flow, glomerular filtration, and liver and kidney size, increase with weight in a nonlinear manner. Based on this allometric theory, the dose in children over 2 years of age could be predicted based on the adult dose using the following relationship:Dose_child_ = Dose_adult_ × (TBW_child_/70)^0.75^
where 70 is the adult reference weight. This formula could also be applied to other measures of body size such as LBW, FFM, and normal fat mass (NFM) [16,89,97].

Thus, it can be expected that Cl, which depends on functional processes such as hepatic metabolism or renal filtration, will scale with an exponent of 0.75. In contrast, structural variables, such as blood volume, vital capacity, the Vd of the central compartment, and steady state, show linear proportionality with body weight and are best fitted with an exponent of 1.0 [89,97].

The equations in a simple allometric compartmental PK model are as follows:Cl_child_ = Cl_adult_ × (TBW_child_/70)^0.75^Vd_child_ = Vd_adult_ × (TBW_child_/70)^1^

We must bear in mind that the allometric model does not take into account the PK changes associated with obesity; however, it reduces the dose per kilogram for larger obese individuals, which is clinically appropriate [89].

### 5.7. Normal Fat Mass

Finally, NFM is an anthropometric descriptor that combines FFM and a fraction of fat mass (Ffat), which can estimate mass by taking body composition into account. It is proposed as a theoretically sound and flexible alternative for adjusting doses in patients of any age and body composition, including obese children and adults [89,97].NFM = FFM + Ffat (TBW − FFM)

Ffat is a fraction that represents how much fat mass contributes to the “effective” pharmacokinetic size, which will depend on each drug and each pharmacokinetic variable such as Vd and Cl. It can have a value of 0 (as for remifentanil and gemcitabine, in which cases the descriptor is equated to FFM), a value of 1 (as for propofol and paracetamol, where the descriptor would be related to TBW), or a value between 0 and 1, which implies a partial contribution of fat mass. If its value is less than zero, this reflects organ dysfunction, as is the case in morbidly obese patients with dexmedetomidine. Drugs such as busulfan have an Ffat of 0.51 for Cl and 0.203 for Vd [89,97].

NFM can be used within an allometric model to predict the relative allometric size compared with standard NFM, which is defined based on an FFM of 56.1 kg for an adult weighing 70 kg and measuring 1.76 m [90].Allometric size: (NFM/NFM_standard_)^3/4^

Thus, Cl in children over 2 years of age can be predicted allometrically from adult estimates accurately and continuously across all ages and weights.Cl_child_ = Cl_adult_ × (NFM_child_/NFM_adult_)^3/4^

This descriptor is not routinely used in clinical practice, although it is being adopted by an increasing number of studies, as it provides a fairly accurate approach to pharmacokinetic changes based on body size and composition and is applicable to a wide range of ages and weights. There is a web tool for calculating busulfan doses based on this theory [89,90,98,99].

### 5.8. Clinical Decision Framework for Body Size Descriptor Selection

Given the lack of verified information on dosing in the obese pediatric patient, the incorporation of PK information and knowledge about pathophysiological alterations is necessary for optimizing doses in different clinical settings.

There is no universal rule for choosing a descriptor that can be applied to all drugs in the context of obesity, since it will depend on their physicochemical and PK characteristics such as liposolubility, plasma protein binding, and hepatic extraction and their metabolism and excretion pathways. In pediatrics, the impact of obesity on drug disposition is superimposed on the age-dependent maturation of absorption, distribution, metabolism, and excretion, so the choice of size descriptor should also take into account the nonlinear changes in body composition and organ function that occur from infancy to adolescence [11,21,22].

We must start with our knowledge of the PK parameters Vd and Cl. Vd is key in establishing regimens requiring a rapid onset of action; Cl is used to establish maintenance regimens to achieve an adequate concentration at steady state.

In selecting the appropriate body size descriptor for a specific drug, two scenarios should be distinguished. Firstly, when the PK parameters in obese patients are known, the choice of descriptor will be made through the comparison with that parameter corrected for weight expressed per kg of TBW, IBW, LBW, or BSA. For example, when calculating the maintenance dose, if the ratio between Cl and TBW is similar between the cohorts of obese and non-obese children, this indicates that excess mass plays an important role in the drug’s clearance, so the descriptor recommended for dose calculation would, in this case, be TBW. If, on the contrary, the ratio is lower in obese patients while it is comparable to the ratio between Cl and IBW, LBW, FFM, or NFM, this implies that the excess mass has little impact on clearance, so it is suggested that doses be based on IBW, LBW, FFM, or NFM, such as, for example, with midazolam or carbamazepine. Nevertheless, in the event that Cl normalized based on weight is not comparable with that of non-obese subjects, it should be plotted graphically against these parameters to establish the exact relationship. A similar rationale can be applied to Vd when selecting the appropriate body size descriptor for the loading dose. It should be taken into account that neither Cl nor Vd increases linearly with body weight; instead, they follow an allometric relationship and are often better correlated with LBW, FFM, NFM, or BSA. Therefore, allometric scaling should be applied, a priori, using an exponent of 0.75 for CL and 1 for Vd and testing descriptors such as FFM, IBW, LBW, NFM, or BSA before TBW [16,57,89,90]. In all these scenarios, the interpretation of weight-normalized clearance should be combined with information on renal and hepatic function, as obesity may be associated with glomerular hyperfiltration or chronic kidney disease, as well as non-alcoholic fatty liver disease and changes in hepatic blood flow and enzyme activity that can modify drug elimination independently of body size [22,23,51,52].

NFM is a descriptor that takes into account IBW or LBW and a fraction of fat mass, a factor that has been described in some drugs, such as, for example, propofol or aminoglycosides. It is presented as a theoretical descriptor that considers mass and composition and can be applied to patients of any age, obese or non-obese. The main drawback that it presents is that the fraction of fat mass is a factor that depends on each drug and PK parameter [89,90].

Secondly, if the pharmacokinetic parameters of CL or Vd for a given drug are unknown in obese children, we have to rely on the drug’s PK characteristics. In this regard, a review was carried out with the aim of developing a decision tool for dosing recommendations for drugs commonly prescribed to obese pediatric patients, selecting the descriptor initially based on the Vd (setting the cutoff point at 5 L/kg) and continuing with the therapeutic margin and its therapeutic window. The authors established an algorithm that is easy to use but that overlooks relevant PK changes that may arise from elimination and excretion processes. However, this approach may oversimplify complex pharmacokinetic processes. For example, vancomycin—characterized by a relatively low volume of distribution in pediatric patients (approximately 0.46–0.56 L/kg) and predominantly eliminated unchanged via glomerular filtration (~75%)—would, according to a purely mechanistic algorithm, be dosed based on IBW. Nevertheless, most pediatric studies with vancomycin have identified TBW, or TBW combined with allometric scaling, as the most appropriate size descriptor for dosing [72,100,101]. In addition, such algorithm-based approaches may fail to adequately reflect drugs with high lipophilicity but limited affinity for adipose tissue. A representative example is digoxin, which has a large volume of distribution (approximately 6–7 L/kg) [100] and a narrow therapeutic index; for this drug, IBW is the recommended dosing descriptor, in contrast to what would be suggested by the algorithm [15].

In our view, this algorithm represents a highly useful approach to dosing drugs for which substantial data gaps exist. However, its clinical applicability could be further enhanced by incorporating, at least for drugs with a low volume of distribution, information on the predominant metabolic pathways and routes of excretion. Likewise, for highly lipophilic drugs, the extent of distribution into adipose tissue relative to other tissues should be considered, as well as the degree of plasma protein binding.

A meta-analysis conducted in adult patients concluded that TBW is the best descriptor for calculating Vd, and IBW is the best descriptor for Cl; likewise, TBW should be used to calculate doses of drugs with moderate or high lipophilicity, and IBW should be used for drugs with low lipophilicity. To date, no study of this nature has been conducted in a pediatric context [30,102].

In Table 4, possible applications of different size descriptors are detailed, based mainly on adult studies, and, for each, several examples in pediatrics are given. We must highlight the knowledge gap in relation to prescribing for obese children. Moreover, most dosing studies conducted in obese pediatric populations are observational in nature, with low-quality evidence and a high risk of bias. To minimize the risk of therapeutic failure or the occurrence of clinically relevant adverse events in obese children, close monitoring of treatment response and potential toxicity is essential, and therapeutic drug monitoring should be employed whenever possible.

## 6. Opportunities and Future Directions

To improve the safety and efficacy of drug treatments for the pediatric population with excess body weight, clinical studies should include more representative pediatric cohorts, covering a wide range of body sizes and always taking into account the ethical issues arising from the inclusion of obese children in clinical trials [30,38]. Another strategy would be to develop prospective randomized studies with different dosage regimens in order to determine the optimal size descriptor for obese and/or overweight children [21,22].

On the other hand, modeling and simulation tools represent an alternative for understanding PK variables in obese children, in addition to promoting the collection of clinical data and fostering close collaboration between healthcare professionals and research institutions [16,30]. The two models most frequently used are those based on population pharmacokinetic data (PopPK) and those that take into account the characteristics of the drug and physiological changes in obese patients (PBPK models).

PopPK analysis is a useful tool for predicting individualized PK parameters (such as AUC, minimum steady-state concentration, Vd, and Cl) using Bayesian estimation based on a specific population model. The advantage of this analysis is that few blood samples are needed for dose adjustment; however, it requires starting from population models with a solid design that provide data across a full range of ages, body sizes, and organ functions.

Simulations based on PopPK analysis allow the most appropriate body size descriptors (TBW, IBW, FFM) to be identified and dosing regimens to be optimized to achieve equivalent exposures in children with and without obesity. In addition, analysis of the probability of achieving the therapeutic target facilitates the selection of a regimen that maximizes efficacy and minimizes the risk of toxicity in this population [16,30].

In this context, it is important to highlight the role of TDM in these patients to adjust plasma concentrations toward a predefined target concentration or therapeutic window using Bayesian approaches, thereby optimizing both dose and dosing interval. TDM is particularly recommended in populations in which variability in ADME processes is not fully characterized, such as obese pediatric patients.

TDM is especially useful for drugs with a narrow therapeutic index and in situations with high inter- and intraindividual pharmacokinetic variability, provided that a clear clinical correlation exists between serum concentrations and pharmacological effects. For appropriate interpretation of TDM results, standardized protocols should be developed to define the types of biological sample, sampling times, and therapeutic ranges. In pediatric practice, TDM recommendations include several drug classes, such as antibiotics (e.g., vancomycin and aminoglycosides), antifungals (e.g., voriconazole and posaconazole), antiepileptics, antipsychotics, immunosuppressants (e.g., tacrolimus and everolimus), and biologics, for which the detection of anti-drug antibodies is also essential to identify loss of response.

Among the limitations of TDM are the lack of rapid and reliable analytical methods for serum concentration determination, uncertainty regarding concentration–response relationships for some drugs, and insufficient evidence to support standardized TDM strategies—an issue that is particularly relevant in the pediatric population [103,104,105].

Physiology-based pharmacokinetic modeling, or PBPK modeling, is a mechanistic approach capable of incorporating the physiological changes associated with obesity to describe alterations in drug disposition in obese children. This type of modeling integrates physiological parameters such as organ size and blood flow, along with specific drug properties (e.g., physicochemical and metabolic characteristics) and study design elements such as doses or sampling times. In this way, PBPK models allow the prediction of pharmacokinetic behaviors and support individualized dosing strategies [16,30,60].

PBPK models have advantages over traditional methods such as PopPK models, as they describe physiological and developmental changes in childhood, allowing the effect of age and body size on drug disposition to be integrated, and they can be used to predict initial PK parameters in obese children and to incorporate mechanistic information that is essential in understanding differences in pharmacokinetic behavior [30,60,106]. However, in clinical practice, TDM based on population pharmacokinetic models is more feasible, as PBPK approaches are more difficult to implement due to the lack of standardized physiological models, the need for specialized software, and the requirement for advanced expertise and training.

PBPK models require detailed physiological information about the study population due to their physiological basis, some of which is still unknown for obese children. Even so, PBPK modeling is a useful tool for simulating pharmacokinetics and exposure for drugs administered in pediatrics, even when the available clinical data are limited [60,107].

This type of study has been conducted in adult patients with obesity to predict the effect of obesity on the Cl of various drugs normalized using different size descriptors [106]. In pediatrics, it has been applied to determine the differences in Vd and Cl between obese and non-obese patients with clindamycin and trimethoprim/sulfamethoxazole [60], and a recent study has applied the principles of PBPK models to assess the impact of obesity on the PK of amlodipine for children and explore the dose adjustment required to achieve the same plasma concentrations as those in non-obese patients [108].

Finally, it is worth mentioning the potential of quantitative systems pharmacology (QSP) models and predictive models based on artificial intelligence (AI) and machine learning (ML) for dose optimization in overweight pediatric patients. QSPs are mechanistic approaches that integrate biological, physiological, and pharmacological data to simulate and predict the response, efficacy, and safety of drugs in complex systems. Combining these models with AI and ML will enable progress in precision medicine and more informed clinical decision-making [109,110].

## 7. Conclusions

This review highlights the fact that the impact of excess body mass in pediatric patients on drug clearance and volume distribution remains insufficiently characterized.

Due to the continuing increase in the prevalence of childhood obesity, a better understanding of how obesity affects PK/PD is a priority in order to optimize dosing regimens for patients with excess body weight.

The extent of these changes in the different PK processes of ADME depends on the properties of each drug, such as lipophilicity and elimination pathways, making it difficult to establish a single dosing strategy for the overweight population.

Given that there are currently no dosing guidelines for obese pediatric patients that are supported by robust clinical evidence, dose adjustments for most drugs must be made with the dual aim of achieving the desired therapeutic effect and minimizing potential adverse effects.

To overcome this gap in knowledge, collaborative efforts between clinicians and researchers should be encouraged in order to develop modeling tools that elucidate alterations in drug disposition. Likewise, joint efforts between pediatricians and clinical pharmacists are needed to standardize dosing guidelines for overweight and obese children, particularly for high-risk drugs and those with a narrow therapeutic index.

## Figures and Tables

**Figure 1 pharmaceuticals-19-00016-f001:**
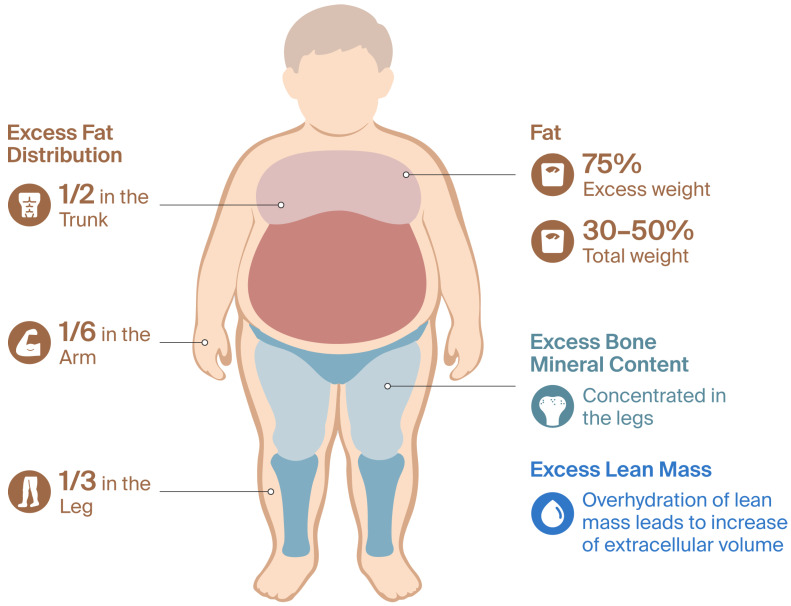
Body composition changes in obese pediatric patients.

**Figure 2 pharmaceuticals-19-00016-f002:**
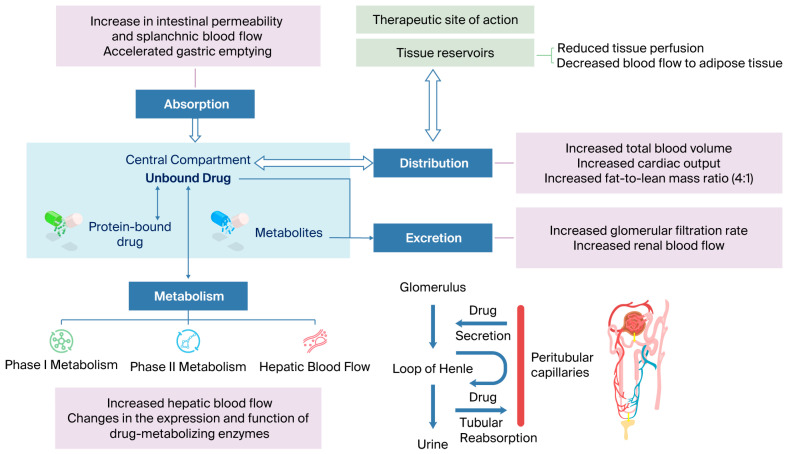
Integrated physiological and PK/PD alterations in pediatric obesity.

**Table 1 pharmaceuticals-19-00016-t001:** Diagnostic criteria for overweight and obesity based on age according to the WHO.

Age Group		2–5 Years	5–18 Years
Index		Weight for Height	BMI
Percentile *	Equivalence Z-Score (Standard Deviation)		
>85th percentile	z-score + 1	Risk of being overweight	Overweight
>97th percentile	z-score + 2	Overweight	Obesity
>99th percentile	z-score + 3	Obesity	Severe obesity

* The 85th, 97th, and 99th percentiles approximate z-scores of +1, +2, and +3, respectively.

**Table 2 pharmaceuticals-19-00016-t002:** Classification of pediatric obesity according to the CDC.

Status	Percentile
Normal weight	BMI > 5th to <85th
Overweight	BMI ≥ 85th to <95th
Class I obesity	BMI ≥ 95th to <120% of the 95th
Class II obesity *	BMI 120–140%
Class III obesity *	BMI ≥ 140%

* According to the American Academy of Pediatrics, severe obesity is classified as Class 2 and Class 3.

**Table 3 pharmaceuticals-19-00016-t003:** Physiological and pathological changes in the obese population and their suggested effects on PK.

Process	Pathophysiological Change	Affected PK Parameter and Expected Change	Observations
Absorption [16,51]	Increased gastric emptying Increased intestinal permeability Increased splanchnic blood flow	F ↔ ka ↑	Clinical relevance of absorption changes should be evaluated in obesity Limited information in pediatrics
Distribution [16,24,30,38]	Increase in total blood volume Increase in cardiac output Changes in organ volume and blood flow Increase in fat mass/lean mass ratio	Vd in low-lipophilicity drugs: slightly ↑ Vd in lipophilic drugs ↑↓ Plasma proteins Albumin ↔ α_1_-acid glycoprotein ↑	Caution with drugs that have a high degree of lipophilicity Limited information in pediatrics
Metabolism [24,30,51,52]	Increased hepatic blood flow Changes in the expression and function of metabolizing enzymes	Cl _hepatic_ Phase I metabolism CYP3A4 ↓ CYP2E1 ↑ CYP2D6 ↑ Xanthine oxidase ↑ Phase II metabolism ↑ Glucuronosyltransferase N-acetyltransferase Glutathione S-transferase	Alteration in hepatic Cl depends on the metabolic pathway Limited information in pediatrics
Excretion [30,51,52,53]	Increased size and renal blood flow	Cl Glomerular filtration ↑ Tubular secretion ↑ Tubular reabsorption ↔	Increase in renal Cl Limited information in pediatrics

Cl: clearance, CYP: cytochrome P; F: bioavailability; ka: absorption rate;; PK: pharmacokinetic; Vd: apparent volume of distribution. Increase: ↑; decrease: ↓; stays the same: ↔.

**Table 4 pharmaceuticals-19-00016-t004:** Body size descriptors and possible applications.

Size Descriptor *	Calculation	Application
Total body weight (kg)	Current patient weight	It could be useful in the dosage of drugs with moderate or high lipophilicity **
Body mass index (kg/m^2^)	Calculation based on growth curves by age and gender Patients aged 2–20 years	To categorize degrees of obesity
Body surface area (m^2^) [81,82]	Haycock’s formula: 0.024265 × TBW (kg)^0.5378^ × height (cm)^0.3964^ Mosteller’s formula: (weight × height/3600)^1/2^	Dosage of antineoplastic agents
Ideal body weight (kg) [83,84,85,86]	Simplified Traub’s formula: IBW = [height in centimeters)^2^ × 1.65] ÷ 1000 Reverse BMI method: IBW = BMI_50_ × [height (m)]^2^ ^#^ Calculation of BMI_50_ according to the 50th percentile of the growth charts by age and sex or Calculation of BMI_50_ according to Callahan and Walker’s formula (see main text)	It could be useful for the dosage of some hydrophilic drugs **
Adjusted body weight (kg)	AjBW = IBW + factor (TBW − IBW) The factor varies between 0.25 and 0.4	It could be useful for the dosage of some drugs that are partially distributed in adipose tissue **
Fat-free mass kg (see main text)/lean body weight (kg) [87,88]	LBW = 3.8 (ECV: extracellular fluid volume) ECV = 0.0215 × TBW^0.6469^ × height^0.7236^ Simplified formula: LBW: IBW + 0.29 (TBW − IBW)	It could be useful for the dosage of some hydrophilic drugs **
Normal fat mass (kg) [89,90]	NFM = FFM + Ffat (TBW − FFM) Ffat is specific to each drug	Theoretical descriptor for calculating doses for a wide range of ages, weights, and body compositions

* We use the term “weight” and/or “mass” based on the commonly accepted concept, although, in practice, we refer to “mass”. ** Data are mainly validated in adults. ^#^ The Centers for Disease Control and Prevention (CDC) charts or the World Health Organization (WHO) tables [29,32].

## Data Availability

No new data were created or analyzed in this study. Data sharing is not applicable to this article.

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
