# Peer review of "Pharmacokinetics and Childhood Obesity: Pathophysiological Basis and Challenges in Choosing the Ideal Body Size Descriptor"

_pharmaceuticals, 2025, doi:10.3390/ph19010016_

Round 1

Reviewer 1 Report

Comments and Suggestions for Authors

In this study, the authors analyze the influence of childhood obesity on the pharmacokinetics and pharmacodynamics of drugs. This review also covers the area of body size descriptors used in clinical practice to optimize dosing for adolescent overweight patients. The introduction initially provides some updates on the necessity; however, it lacks the existing reports and background information on the work. I would like to suggest the following:

  1. The study needs a proper graphical abstract for a clear understanding of the purpose of this report.
  2. The introduction should be more detailed, highlighting the previous reports and the justification of the current report. In line 79, the previous reports 14~22 could provide sufficient background.
  3. Line 115 and 117, Table. 1 was repeated. As per the WHO, the diagnostic criteria are BMI-based and considered overweight or obese if the weight-for-height is greater than multiple standard deviations above the WHO Child Growth Standards median. Table 1 can be further rearranged with more information.
  4. 2: Because all these are values, “to” in between the ranges is suitable. The CDC (BMI category and BMI ranges) table is easier to read, and therfore authors are suggested to simplify the ranges to numbers only. Also, the American Academy of Pediatrics classifies severe obesity as Class 2, and Class 3 will be a good piece of information for the table footnote.
  5. Table 3: A Separate description text for these symbols will be needed. In section 4, some description can be added. Authors should explain why the pathologic change in obesity has no “clinical relevance” on absorption. Also, an appropriate reference for Table 3 will be suggested.
  6. It would be benificial to make a list of scientific cases for explaining the pathological change in obesity. Examples like clindamycin and trimethoprim/sulfamethoxazole in obese and non-obese adult patients are also acceptable.
  7. Line 365, Other phase II metabolic enzymes can be added to another subsection.
  8. Please correct the reference position in line 413.
  9. For readers’ convenience, rather than a bunch of references in line 487, a separate reference for each descriptor is recommended.
  10. Please confirm that the calculation A~E in Table 4 is presented accurately. The PCT at the bottom of the table was not abbreviated properly.

Reviewer 2 Report

Comments and Suggestions for Authors

Dear Authors,

Thank you for submitting your manuscript on the physiopathological basis and challenges of drug dosing in children with obesity. This narrative review addresses an important and increasingly relevant topic, and the structure—from conceptualisation of obesity to physiological changes, pharmacokinetic/pharmacodynamic (PK/PD) implications, size descriptors, and future perspectives—is logical and clearly developed. The manuscript is generally well written and provides a valuable overview of the field. However, in its current form, the review would benefit from further refinement to improve clarity, completeness, and practical applicability. Below, I outline suggestions divided into content-related andformatting/presentation-relatedconsiderations.

Content Suggestions

  1. Administration Routes: The review discusses oral absorption but does not address potential changes in intramuscular (IM), subcutaneous (SC), or rectal absorption in children with obesity. Including these aspects would strengthen the comprehensiveness of the PK section.
  2. Use of TBW vs. TDM (Lines 253–257): In this section, it would be beneficial to specify clinical situations in which total body weight (TBW) or therapeutic drug monitoring (TDM) is preferred, especially when alternative dosing rules exist. It would also be helpful to clarify whether using TBW results in decreased effectiveness or increased risk.
  3. Role of TDM: Given the uncertainties in dose selection in paediatric obesity, a dedicated section on the role, value, and limitations of TDM in this population would be highly relevant.
  4. Clearance and LBW (Lines 278–282): The paragraph on clearance being determined by lean body weight (LBW) is descriptive but does not provide interpretation. It would be helpful to assess the robustness and limitations of the available evidence and summarise clinical implications.
  5. Safety of Paracetamol: Is there evidence that obese children experience higher rates of toxicity with paracetamol? If so, this should be briefly discussed, particularly regarding altered metabolic pathways and formation of toxic metabolites.
  6. Estimation of GFR: Recommendations regarding the most appropriate glomerular filtration rate estimation method would be valuable. Should clinicians use drug-specific equations (e.g., vancomycin-adjusted models) or apply formulas such as LMR18?
  7. Chapter Title Adjustment: Since the section discusses both size and body composition descriptors, renaming Chapter 5 to “Body Size and Composition Descriptors” would improve accuracy.
  8. Future Directions: The section on future perspectives emphasises physiologically based PK (PBPK), but appears to undervalue population pharmacokinetics (PopPK) and Bayesian estimation. As TDM strategies frequently rely on these approaches, and they often require fewer samples, it would be appropriate to reflect this benefit.

Presentation and Formatting Suggestions

  1. Including one or more figures summarising key physiological and PK–PD alterations would significantly enhance readability.
  2. The list of abbreviations requires review. Several acronyms are missing or inaccurately defined (e.g., PoPK should be PopPK).
  3. Line 115 contains inconsistent font size and should be corrected.
  4. Please standardise the formatting of “Table” (bold vs. non-bold) throughout the text.
  5. In scientific notation, Ka should use a lowercase ‘k’.
  6. In Table 3, arrows vary in thickness. If this variation indicates magnitude of change, please clarify this in the table legend; otherwise, standardise arrow formatting.

Overall, this is an interesting, timely, and well-structured review. With the suggested improvements—particularly strengthening practical conclusions and addressing missing content—the manuscript will become more useful for clinicians and researchers working in paediatric pharmacotherapy.

Congratulations on the work completed so far, and I hope my comments support further refinement.

Reviewer 3 Report

Comments and Suggestions for Authors

pharmaceuticals-4022026

Pharmacokinetics and childhood obesity: pathophysiological basis and challenges in choosing the ideal body size descriptor

The manuscript addresses an important and clinically relevant topic: pharmacokinetic and pharmacodynamic alterations in obese pediatric patients and the difficulties in selecting appropriate body size descriptors for drug dosing. The manuscript has potential value; however, several issues related to structure, clarity, and completeness limit its current impact. Substantial revisions are needed.

  1. The manuscript contains many extremely short paragraphs (1–3 sentences), which disrupt the flow and weaken the presentation of key points. Please combine related ideas into more cohesive paragraphs and reorganize sections so that each paragraph develops a clear, unified concept. This is a major concern, as the fragmentation prevents the manuscript from emphasizing the central messages of the review.
  2. The Introduction should explain why special attention is needed for children with obesity, including how pharmacokinetic changes differ between obese adults and obese children, what developmental physiological factors make pediatric dosing more complex, and why existing adult-based models or size descriptors are insufficient for pediatric populations.
  3. Table 4 is overly detailed due to inclusion of multiple equations. Please move all equations into the main text, where they can be explained more clearly, and keep the table concise by summarizing the key descriptors only.
  4. I strongly recommend adding figures to support and illustrate key concepts.

Round 2

Reviewer 1 Report

Comments and Suggestions for Authors Thank you for sharing the revised manuscript. I appreciate the author’s effort in revising the manuscript for readers.  The current version has been significantly corrected and improved to better meet readers’ expectations.   I have some minor comments: Figures 1 and 2, newly incorporated, which improve the content of the revision. The graphical abstract was not included in the revised submission; therefore, I will suggest modifying Figure 1 with a comparative healthy body composition side-by-side to present in the graphical abstract, unless the authors have already prepared something.   In the Phase 1 metabolism section, CYP3A4, CYP2E1, and Other cytochrome CYP450 enzymes can be presented in subsections, for example, 4.3.1.1.~3.  

Author Response

Comments and Suggestions: Thank you for sharing the revised manuscript. I appreciate the author’s effort in revising the manuscript for readers.  The current version has been significantly corrected and improved to better meet readers’ expectations.   I have some minor comments: Figures 1 and 2, newly incorporated, which improve the content of the revision. The graphical abstract was not included in the revised submission; therefore, I will suggest modifying Figure 1 with a comparative healthy body composition side-by-side to present in the graphical abstract, unless the authors have already prepared something.   In the Phase 1 metabolism section, CYP3A4, CYP2E1, and Other cytochrome CYP450 enzymes can be presented in subsections, for example, 4.3.1.1.~3.

Response: 

Thank you very much for your valuable comments and suggestions, which we believe have undoubtedly enriched the manuscript.

Regarding the graphical abstract, we have already uploaded the final version to the submission platform, in which we have included the objectives, figures detailing the results, and the conclusions. We hope that it adequately addresses your comments.

We have taken into consideration your suggestion regarding the subsection on metabolic enzymes and have detailed the indicated subsections (see lines 362, 375, and 404).

Reviewer 2 Report

Comments and Suggestions for Authors

Congratulations

Author Response

On behalf of all the authors and myself, I would like to sincerely thank you for the valuable comments and suggestions provided, which we believe have undoubtedly improved and enriched the manuscript.

Reviewer 3 Report

Comments and Suggestions for Authors

The manuscript was appropriately revised. However, the authors must ensure that all figures are original. If any figures are adapted from previously published sources, please include proper copyright permissions.

Author Response

Comments and Suggestions for Authors: The manuscript was appropriately revised. However, the authors must ensure that all figures are original. If any figures are adapted from previously published sources, please include proper copyright permissions.

Response: 

We understand and appreciate the revisor concern regarding the figures included in the revised manuscript. We would like to confirm that Figure 1 and Figure 2 are original figures created by the authors, based on a basic ADME scheme.